# Can We Assume the Gene Expression Profile as a Proxy for Signaling Network Activity?

**DOI:** 10.3390/biom10060850

**Published:** 2020-06-03

**Authors:** Mehran Piran, Reza Karbalaei, Mehrdad Piran, Jehad Aldahdooh, Mehdi Mirzaie, Naser Ansari-Pour, Jing Tang, Mohieddin Jafari

**Affiliations:** 1Bioinformatics and Computational Biology Research Center, Shiraz University of Medical Sciences, Shiraz P.O. Box 71336-54361, Iran; piranmehran@gmail.com; 2Department of Biology, Temple University, Philadelphia, PA 19122, USA; reza.karbalaei@temple.edu; 3Department of Tissue Engineering and Applied Cell Sciences, School of Advanced Technologies in Medicine, Shahid Beheshti University of Medical Sciences, Tehran P.O. Box 14177-55469, Iran; mehrdadp1989@gmail.com; 4Research Program in Systems Oncology, Faculty of Medicine, University of Helsinki, 00270 Helsinki, Finland; jehad.aldahdooh@helsinki.fi; 5Department of Applied Mathematics, Faculty of Mathematical Sciences, Tarbiat Modares University, Tehran P.O. Box 14115-134, Iran; mirzaie@modares.ac.ir; 6Big Data Institute, Nuffield Department of Medicine, University of Oxford, Oxford OX3 7LF, UK; naser.ansari-pour@bdi.ox.ac.uk

**Keywords:** gene expression, signaling network, network biology, transcriptomics, differentially expressed genes, causality analysis

## Abstract

Studying relationships among gene products by expression profile analysis is a common approach in systems biology. Many studies have generalized the outcomes to the different levels of central dogma information flow and assumed a correlation of transcript and protein expression levels. However, the relation between the various types of interaction (i.e., activation and inhibition) of gene products to their expression profiles has not been widely studied. In fact, looking for any perturbation according to differentially expressed genes is the common approach, while analyzing the effects of altered expression on the activity of signaling pathways is often ignored. In this study, we examine whether significant changes in gene expression necessarily lead to dysregulated signaling pathways. Using four commonly used and comprehensive databases, we extracted all relevant gene expression data and all relationships among directly linked gene pairs. We aimed to evaluate the ratio of coherency or sign consistency between the expression level as well as the causal relationships among the gene pairs. Through a comparison with random unconnected gene pairs, we illustrate that the signaling network is incoherent, and inconsistent with the recorded expression profile. Finally, we demonstrate that, to infer perturbed signaling pathways, we need to consider the type of relationships in addition to gene-product expression data, especially at the transcript level. We assert that identifying enriched biological processes via differentially expressed genes is limited when attempting to infer dysregulated pathways.

## 1. Introduction

In network biology, defining causal relationships among nodes is crucial for static and dynamic analysis [1,2]. The most available high-throughput data for inferring molecular relationships are arguably whole-transcriptome expression profiles analyzed with statistical models [3]. The challenge is extrapolating causality in signaling and regulatory mechanisms from a significant correlation between any given gene pair. Many spurious correlations among gene pairs lacking any direct causal relationship are possible, both stochastically and as a result of indirect effects [4]. So far, reverse engineering algorithms have been developed to tackle this challenge and to infer gene networks and regulatory interactions from expression profiles [5,6,7].

When considering signaling networks as a portrait of causal relationships among the molecular entities in biology, their leading players are proteins, whose activity is often regulated by post-translational modifications such as phosphorylation and acetylation [8,9]. Therefore, inference of signaling networks can be directly inferred from (Phospho) proteomic and protein–protein interaction data [10]. However, these kinds of data are expensive and laborious to acquire. Given the correlation between protein and gene expression, a common alternative approach is using gene expression to infer interactions between proteins. It is known that gene expression or transcriptome refers to “what appears to happen in a biological system”, while the signaling network explicates “what makes it happen and what has happened in a complex view of the system” [11]. Meanwhile, in transcriptomic studies, differentially expressed genes (DEGs) are identified based on the changes in mRNA levels between two or more groups of samples, and several affected biological pathways are recognized by node-centric enrichment analysis [12,13]. This, therefore, begs the question of whether gene expression profiles amplify the mechanism of signaling circuits, i.e., activatory/inhibitory relationships.

In this study, we aimed to examine the coherency between expression profiles and the signs of relationship in signaling networks for all possible gene pairs. To elaborate on what we mean by coherency, imagine a gene pair (*G_I_, G_II_*) where gene *G_I_* activates gene *G_II_*. If the expression profiles of both are correlated positively, we infer that expression data strengthen the activation type of this signaling relationship and are thus a coherent gene pair. In contrast, let gene *G_I_* inhibit gene *G_II_*. In this case, the coherent gene pairs are negatively correlated, and the inhibition type of this signaling relationship is strengthened in accordance with the expression levels of transcripts. If gene *G_I_* activates gene *G_II_* while there is a negative correlation between them or if gene *G_I_* inhibits gene *G_II_* while there is a positive correlation between them, this implies the incoherency between the gene pair relationships. In other words, incoherent gene pairs’ effects on the action of each other are not supported by their mRNA expression levels. In addition to these simple scenarios, we also considered more complicated subgraphs in a signaling network (See Table 1). Edges appear between gene pairs, indicating a relationship. Since all edges have a direction, the type of logical relationship is determined using the sign of edges. We then performed a correlation analysis between the connected gene pairs compared with unconnected gene pairs to infer coherency. To this end, we used expression datasets in the Gene Expression Omnibus (GEO) [14] and Genomics of Drug Sensitivity in Cancer (GDSC) databases [15] to extract the relevant gene expression profiles. Two literature-curated databases for signaling pathways, namely the Kyoto Encyclopedia of Genes and Genomes (KEGG) [16] and OmniPath [17] (which integrates literature-curated human signaling pathways from 34 resources) were used to extract the sign of relationships among directly linked gene pairs. Therefore, a coherency analysis was undertaken independently for all four combinations of databases in parallel (see Table 1). This analysis allowed us to explore the importance of having coherent gene pair relationships to infer the dysregulated signaling pathways.

## 2. Materials and Methods

In this study, four independent analyses were performed based on two gene expression databases, i.e., GEO and GDSC, and two signaling pathway databases, i.e., KEGG and OmniPath, in parallel (Figure 1) [14,15,16,17]. The signaling pathway databases were thus independently used to reconstruct a whole signaling network, and the gene expression databases were separately used to apply a correlation analysis to each gene pair in the pathways to undertake and compare the distinct analyses and findings of KEGG/GEO, KEGG/GDSC, OmniPath/GEO, and OmniPath/GDSC. To briefly introduce the utilized gene expression databases, GEO is an international public repository of National Center for Biotechnology Information (NCBI) that archives microarray and next-generation sequencing expression data. The GDSC database is the largest public repository that archives information about drug sensitivity in cancer cells and biomarkers of drug response in these cells. In this work, gene expression profiles from GDSC cell lines and GEO studies were used to extract pairwise association between genes.

### 2.1. Signaling Network Reconstruction

Here, we focused on human signaling pathways based on available datasets. All human-related signaling pathways were downloaded from the KEGG database. Using the KEGGgraph package [18], these pathways were imported into the R environment [19]. Edge information was extracted, and each graph was converted to an edge list. Next, all edges (*n* = 26,490) were merged, and a directed signed signaling network was reconstructed (Appendix A). Eligible edges (see Section 2.3) were then selected, and correlation analysis was undertaken on eligible gene pairs. The pypath python module [17] was also used to do the same and create an edge list based on the OmniPath database (see Appendix A). This edge list (*n* = 20,853) was also imported into the R environment for the downstream statistical analysis on the gene pairs.

### 2.2. Gene Expression Profiles Extraction

The standard GEO query format (GEO Profiles) was used to identify all up- and down-expressed genes or DEGs represented within the KEGG and/or OmniPath edge lists. For each study, a number of differentially expressed genes are identified based on the desired thresholds of log fold change and *p*-values indicating significance. As the up- and downregulated gene expression profiles are derived from different kinds of studies corresponding to different periods and different genomic technologies, we relied on the definition of up- and downregulated genes in the GEO database to define DEGs. Gene expression profiles available in GDSC were downloaded for both edge lists, followed by preprocessing and outlier detection. Finally, based on GEO and GDCS, four expression matrices were created using the genes which make up of KEGG and OmniPath edge list (Appendix A). For more details about the total number of gene expression profiles, DEGs, and total number of samples that were downloaded from GEO and GDSC, see Figure 1 and Table 2.

### 2.3. Mutual Association Analysis

In the next step, the correlation of the expression profiles of each gene pair was statistically tested. For computing correlation coefficients for any gene pair, including Pearson, Spearman’s rank, and Kendall rank coefficients, we only considered gene pairs having more than two samples. These gene pairs were considered as eligible edges for downstream statistical analysis (Appendix A). Samples with expression data for the gene pairs may have come from different datasets with multiple organisms and tissues and therefore they were analyzed separately and independently. Figure 2A represents the effect of this preprocessing on an exemplary gene pair in our dataset. In this study, the gene expression profiles were considered dataset-specific, as mentioned, to avoid any inconsistency among the samples collected from diverse datasets, as sample heterogeneity can easily obscure pairwise relationships. An edge is therefore considered homogeneous if the correlation sign is consistent throughout. These homogeneous edges were used for correlation analysis. Then, according to the statistical significance and the sign of the correlation coefficient, the coherent and incoherent gene pairs were inferred (Appendix A). Note that a *p*-value less than 0.05 was considered to represent the cutoff.

### 2.4. Randomly Selected Unconnected Gene Pairs

The edge lists obtained in the previous step were converted into adjacency matrices using the igraph package in R [20]. Then, the adjacency matrix was self-multiplied more than the diameter of the network (e.g., *n* > 17) (Table 2). We then randomly selected 1000 unconnected gene pairs ten times, for which the corresponding elements in the matrix were zero (gene pairs with no direct immediate and non-immediate interactions). For these gene pairs, which we called unconnected gene pairs (UGPs), the same downstream analyses, i.e., pre-processing and correlation analysis, were implemented to compare the significance and sign of correlation coefficients to connected gene pairs (Appendix A).

### 2.5. Complex Subgraphs

Since gene pairs are not isolated within the whole signaling networks, the larger subnetworks, which consisted of gene pair relationships, needed to be considered. Therefore, we extracted specific subgraphs from the signaling networks to investigate any association between gene expression profiles and the complex structures of relationships for each gene pair. DNFBL (Dual Negative Feedback Loop), DPFBL1 (Dual Positive Feedback Loop 1), and DPFBL2 (Dual Positive Feedback Loop 2) are subgraphs of gene pairs that influence each other directly twice (see Table 1 for the full names of subgraphs). These pairs are readily found by checking the source and target nodes in the edge lists (or upper and lower triangles in adjacency matrices). We then focused on the connected gene pairs, which also influence each other indirectly by a sequence of intermediate nodes. Following matrix self-multiplication, the weighted and unweighted adjacency matrices of the giant component of eligible edges in the signaling network were powered by the network radius magnitude. Considering that the network is directed and the adjacency matrix is not symmetric, the feed-forward and feedback loops, i.e., MNFBL1-2, MPFBL1-2, MFFL1-2, and MNFFL1-2, were determined (Table 1). For a more detailed explanation of procedures, see Appendix A.

## 3. Results

The overall details of the four parallel coherency analyses are presented, including the dimension of the expression matrices generated from whole-transcriptome expression profiles, and the size and diameter of the giant component in each analysis (Table 2). Notably, the number of DEGs was higher in OmniPath than KEGG, even though the size of the KEGG network is 1.8-fold larger than the OmniPath network. The ratio of eligible edges to all edges was calculated for all four analyses (see Figure 2B). The ratio of eligible edges in the OmniPath edge list also was higher than KEGG based on both GDSC and GEO databases. In addition, the ratio of eligible edges was higher in GDSC compared with GEO. This comparison may indicate the higher quality of gathered and annotated data in OmniPath and GDSC databases.

### 3.1. The Ratio of Coherency for Gene Pairs

After filtering out heterogeneous edges, an extensive list of homogeneous edges was constructed (Appendix A) and Supplementary File 6) for correlation analysis. We performed correlation analysis in datasets with more than two samples, and the majority of the datasets had more than five samples. The violin plots of Pearson correlation coefficients for each analysis are shown in Figure 3A. The distribution of the coefficients showed a nearly uniform distribution with a little left skewness for KEGG/GEO and OmniPath/GEO, while for KEGG/GDSC and OmniPath/GDSC, it followed a normal distribution with the median at approximately zero. In addition to the issue of different sample sizes in GEO and GDSC, this suggests that for GDSC-based edges, correlations between the expression profiles of the gene pairs do not tend to show a high positive or negative correlation. In other words, for a given gene pair (I, II), over- or under-expression of gene I does not have a substantial effect on the expression of gene II regardless of the edge sign.

Figure 3B depicts the ratios of coherent and incoherent gene pairs along with the number of non-significant edges that have False Discovery Rate (FDR) adjusted *p*-values larger than 0.05, and we could not establish the coherency status at a likelihood greater than or equal to 95%. In addition, the sum of the ratio of incoherent gene pairs and non-significant edges was more than the ratio of coherent gene pairs in all four analyses. The ratio of coherent gene pairs in OmniPath was, in general, higher than for KEGG, while the ratio of coherent gene pairs in the GDSC database was higher than that of GEO. In more detail, when we considered GDSC as a cancer-specific homogenous dataset, the percent of coherent edges increased from 13.5% in GEO to 28.1% according to the KEGG database and from 11.1% in GEO to 34.2% based on the OmniPath database. In the following figures, the results correspond to the Pearson correlation coefficient. However, note that the ratio of coherency for gene pairs is also similar when using Spearman and Kendall rank correlation coefficients (Appendix A).

Figure 4 shows the FDR-adjusted *p*-values versus correlation coefficients of activation and inhibition edges in all four analyses. The symmetric pattern of coefficients is recognizable for both activation and inhibition edges in the four analyses. This suggests that the correlation between a given gene pair is not predominantly affected by the sign of the interaction. In other words, although activation edges illustrate an overrepresentation of strongly positively correlated gene pairs in all four analyses, the inhibition edges do not display any enrichment in the strong negative side of plots compared with the strong positive side. This also demonstrates that the majority of coherent gene pairs are activation rather than inhibition edges. In the next step, we tried to explore and provide supporting evidence for the incoherent gene pairs by focusing on more complex subgraphs in the signaling network.

### 3.2. The Ratio of Coherency on Subgraphs

In this step, we explored whether complex subgraphs are coherent compared to when only single edges are considered. Briefly, we assumed that observing some incoherency of activation and inhibition edges may depend on the complex structure of the signaling network and, thus, the behavior of larger subgraphs should be considered to infer coherency (Figure 5). Like the simple activation or inhibition edges, correlations are computed and categorized by taking into account the correlation sign for each mentioned subgraph and the calculated FDR-adjusted *p*-values (see details in Appendix A for all the four analyses). For example, we expected that the proportion of significant positive correlations is more than the negative correlations in DNFBL as a negative feedback loop when compared with DPFBL1 and DPFBL2, because the two edges of the DNFBL do not have the same sign, and overexpression of one protein entails the under-expression of the other one in a negative feedback loop. However, this expectation was only partially fulfilled in KEGG/GEO analysis.

To statistically compare the coherency of different subgraphs, the correlation analysis was also implemented on multiple sets of 1000 randomly unconnected gene pairs (UGPs), and a binomial proportion test was then undertaken to compare all of the proportions as illustrated in Figure 5. UGP sets were employed to compare the results of the coherency analysis with the ones obtained from gene pairs where relationships exist between them. In most cases (>60%), the comparison showed a statistically significant difference between pairwise proportions of UGPs and the connected gene pairs, e.g., Act, Inh, DNFBL for all four combinations of databases (Appendix A). It is possible that the connected genes are affected by each other with respect to UGPs, but it may happen in a more complex way that it is not inferred by correlation analysis (Figure 5). We also aimed to continue our search to check coherency in larger subgraph structures. We therefore identified subgraphs that contained more than two edges, i.e., MNFBLs (Multiple Negative Feedback Loop 1 and 2), MPFBLs (Multiple Positive Feedback Loop 1 and 2), MFFLs (Multiple Feed-Forward Loop 1 and 2), and MNFFLs (Multiple Negative Feed-Forward Loop 1 and 2), using matrix self-multiplication (See Table 1). The purpose of self-multiplication was to retrieve the related gene pairs in complex subgraphs and unrelated gene pairs in unconnected gene pairs. However, our results demonstrated the limitation of strong coherent relationships among gene pairs, suggesting that, independent of the structure of the subgraph, gene expression profiles are discordant with the functionality of signaling circuits.

## 4. Discussion

Incoherency in signaling networks at the transcript level might be a strategy used by cells to tune both genetic and environmental signals or a byproduct resulting from frequent modulation of signal transduction within biological systems. In fact, cells acquire the characteristics they need by setting different layers of gene regulations, which illustrates the importance of different layers of post transcriptional regulation [21]. High-throughput technologies, such as mRNA microarray, CHIP-seq (Chromatin Immunoprecipitation Sequencing), and mass spectrometry proteomics, have uncovered the amount of expression through determination of mRNA and protein levels [22]. There is an apparent correspondence between mRNA and protein concentrations [23]. Nonetheless, more than fifty percent of protein variation cannot be explained by variation in mRNA concentration [23,24,25]. These unexplained variations might come from organism-specific translational and post-translational regulations, including protein degradation by ubiquitylation and sumoylation [26]. Because we were dealing with whole-human signaling networks, considering all aspects that impact the expression of each gene is far-fetched. Such an analysis would represent the total coherency based on the analysis of gene expression at the mRNA level. The correlation between transcript and protein concentrations are considerable for some house-keeping genes. However, in many eukaryotes, there is no strong correlation for genes of signal transduction or transcriptional regulation, while their encoded proteins are often involved in different signaling networks and determine cell fate and behavior of the system [27].

Although the regulation of gene expression results in a particular concentration of proteins, it is not sufficient to completely describe protein abundances [28,29]. The roles of other mechanisms, such as post-transcriptional, post-translational, and protein degradation regulations, have been reported to control steady-state protein abundance and protein activity [28,30]. For example, miRNAs are associated with proteins and diseases as regulators of gene expression [31,32]. There are several studies focusing on the regulatory role of long noncoding RNAs (lncRNAs) in wound healing and cancers [33,34,35]. Alternative splicing also results in diverse forms of proteins, and the functions of proteins and corresponding signal transduction pathways are affected by this type of regulation in some diseases [36]. Moreover, numerous methylation sites have been discovered on human proteins that impact cell signaling [37]. Therefore, the incoherency might come from the effects of complicated regulatory networks to buffer signaling networks within biological systems. These modifications are thus likely to be the source of poor coherency between signal transduction and gene expression and are particularly valid for multicellular eukaryotes such as worm and fly. In contrast, yeast genes engaged in signal transduction exhibit high correlation between mRNA and protein concentrations [24]. Interestingly, Larsen et al. recently demonstrated that there is no causal relationship between the expression of transcription factors and their targets in the gene regulatory network of *Escherichia coli* and, therefore, the transcriptional regulation cannot be adequately addressed by examining the current static gene regulatory networks [38]. As a result, inferring gene regulatory or signaling relationships from transcript data is challenging because these data are not the proxy of molecular activity. Only in some cases are the results acceptable for constructing logical circuits of biological elements, e.g., if the components of the system are all kinases and the transition of the signals is related to the phosphorylation process [10,39,40]. Different layers of gene regulation need to be considered for each gene pair when inferring causal relationships between a pair of genes. Previously, we also demonstrated the association of altered expression with the signaling circuits in chronic obstructive pulmonary disease [41] and rabies infection [42] as case studies.

We implemented three different types of correlation analyses, namely Pearson, Spearman, and Kendall, all of which exhibited similar findings. Based on the correlation results in Figure 3A, the volcano plots in Figure 4, which exhibit no significant difference between activation and inhibition edges, and the ratios in Figure 5, causal relationship can be inferred poorly at the transcript level, at least in multicellular eukaryotic organisms such as *Homo sapiens*. Proportional binomial tests suggested that there are statistically significant differences between UGPs and other subgraphs (Appendix A), and this demonstrates that the structure of subgraphs affects the coherency. Based on these considerations, it is therefore strongly advocated that information in signaling networks be used or that relationships between the genes are defined in addition to assessing gene expression at both transcript and protein levels.

## 5. Conclusions

Although there is a general assumption that the expression level could strengthen or weaken the signal to transduce in the signaling pathways, we show that the mRNA level of gene pairs is generally incoherent with the way they manipulate one another (Figure 5). We also showed that there is a level of association between the structure of the subgraphs and gene pair expression profiles. Expression profiles of the unconnected gene pairs were statistically more independent than connected ones. Note that these observations are according to the four datasets used here, and independent investigation of gene pair relationships based on more accurate and specific datasets is recommended.

In this study, we aimed to focus on the impact of the type of relationship on any given stimulated signaling pathway, an area which is usually ignored in functional genomic studies. We demonstrate that DEGs have only partial information on the whole story of the associated mechanism. The presented findings support the idea of using interaction- or edge-centric (based on DEG–DEG activatory or inhibitory relationships) enrichment analysis against node-centric (based only on a list of DEGs) analyses to provide a more vivid perspective of implicated pathways. This is because the majority of the altered expression of DEGs gradually disappears and is then overlooked by the whole signaling network system, whether stimulated endogenously or exogenously.

## Figures and Tables

**Figure 1 biomolecules-10-00850-f001:**
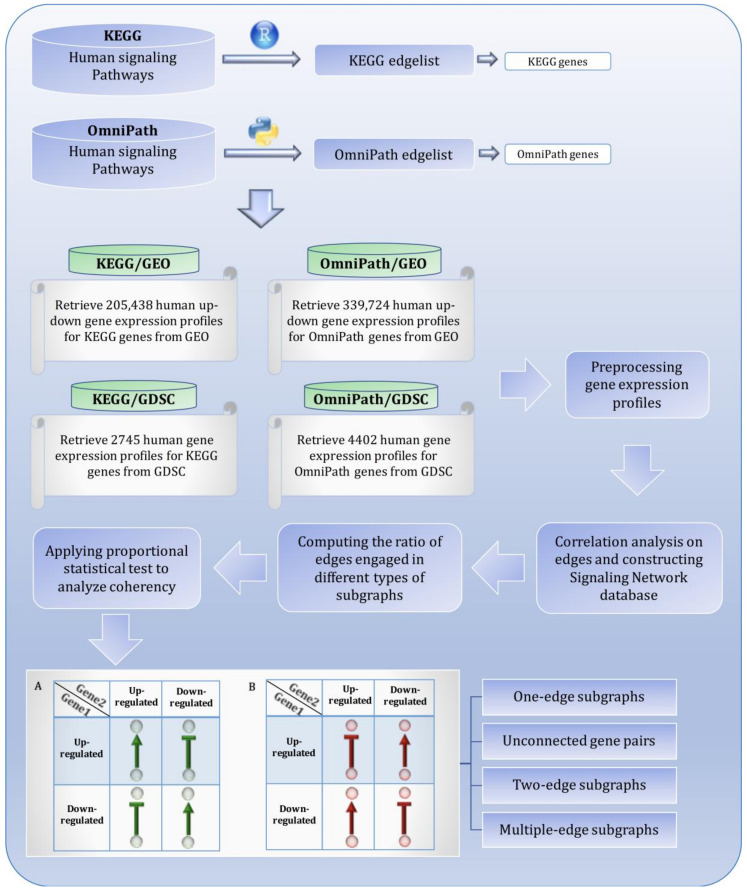
Visual overview of how information from different databases was integrated to analyze the coherency. An edge list was constructed from Kyoto Encyclopedia of Genes and Genomes (KEGG) and OmniPath databases. All the gene expression profiles for the edge list genes were then downloaded from Gene Expression Omnibus (GEO) and Genomics of Drug Sensitivity in Cancer (GDSC) databases. Next, data were preprocessed, and a suitable structure was created for correlation analysis among the gene pairs. By interpreting the information from correlation tests and the proportional tests, coherency analysis was implemented on different forms of subgraphs. There are a total of four coherent conditions in panel A and four incoherent conditions in panel B. For instance, in panel A, if gene1 is up-regulated and there is an activation between the gene pair, gene2 must be up-regulated. In panel B, if gene1 is up-regulated and there is an inhibitory relationship between the gene pair, gene2 is expected to be up-regulated.

**Figure 2 biomolecules-10-00850-f002:**
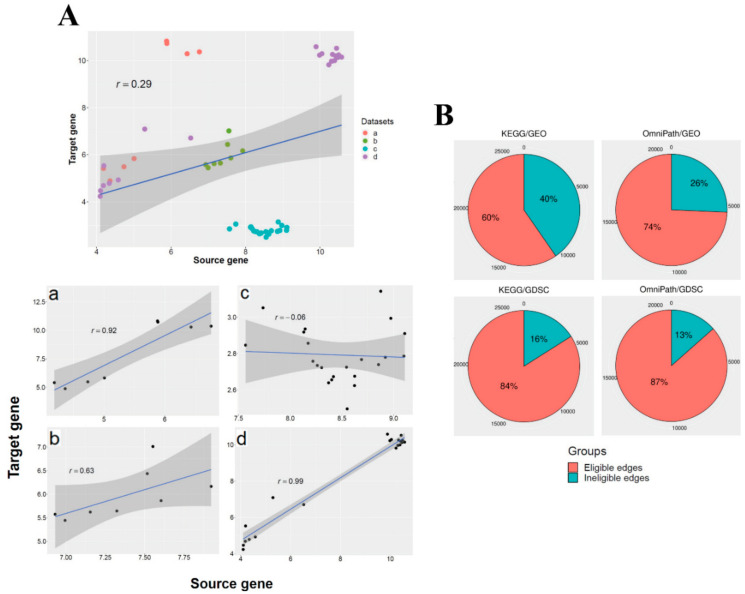
(**A**) An exemplary relationship between gene pair expression. These scatter plots contain the Pearson coefficient correlations and fitted linear regression line. The X-axis and Y-axis values differ according to the expression profile of this gene pair in different gene expression datasets. The gene expression profiles of these exemplary gene pairs in the edge list before pre-processing are depicted. The same gene pairs’ expression profiles were separated according to the four relevant datasets. (**B**) The proportion of eligible and ineligible edges in the four parallel analyses. The numbers around each chart represent the number of edges at that point.

**Figure 3 biomolecules-10-00850-f003:**
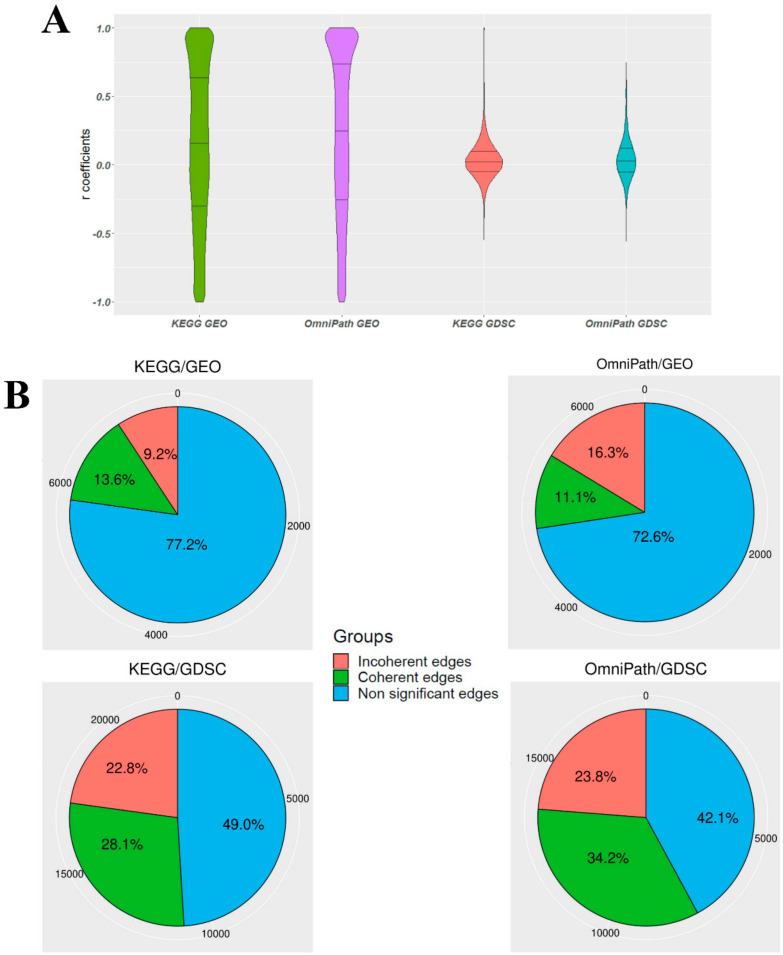
(**A**) Distribution of Pearson correlation coefficient values for the four parallel coherency analyses. (**B**) The ratio of coherent and incoherent edges in addition to the ratio of non-significant edges that have adjusted *p*-values of correlation test larger than 0.05. The values around each pie chart represent the exact numbers.

**Figure 4 biomolecules-10-00850-f004:**
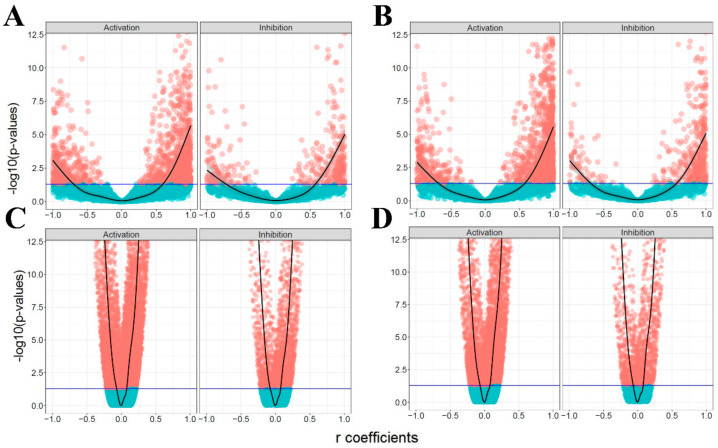
The volcano plots of the activation and inhibition edges. The horizontal axis is the Pearson correlation coefficient, and the vertical axis shows log transformed False Discovery Rate (FDR)-adjusted *p*-values. The threshold line (blue) represents the significance cut-off value of 0.05. (**A**–**D**) plots correspond to KEGG/GEO, OmniPath/GEO, KEGG/GDSC, and OmniPath/GDSC analyses, respectively.

**Figure 5 biomolecules-10-00850-f005:**
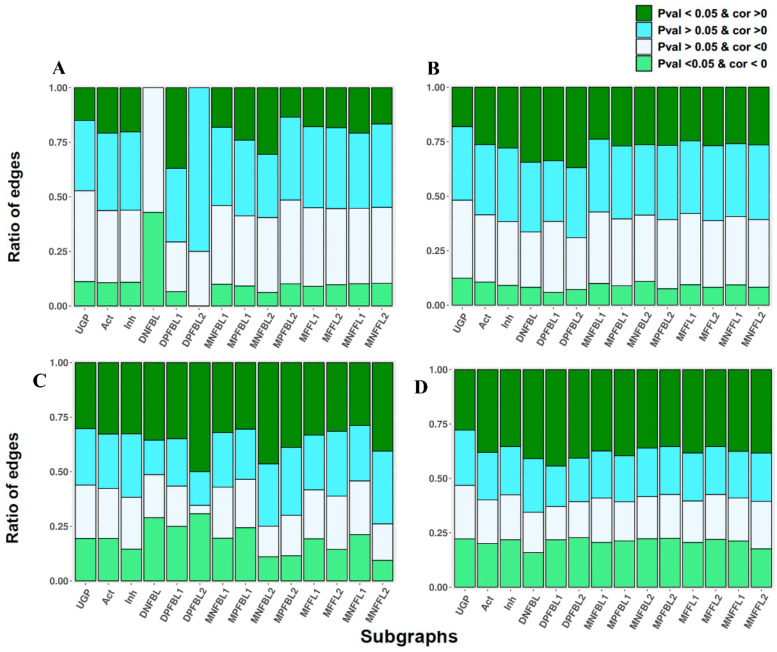
The ratio of eligible and homogeneous edges involved in different subgraphs are represented by stacker bar plots for all four analyses. (**A**,**B**) are KEGG/GEO and OmniPath/GEO plots, and (**C**,**D**) plots correspond to KEGG/GDSC and OmniPath/GDSC. Note that the unconnected gene pair (UGP) ratios are the average ratio over of all UGP sets. See Table 1 for the full names of subgraphs.

**Table 1 biomolecules-10-00850-t001:** Details of different subgraphs present in all biological signaling networks. The dashed lines indicate multiple edges between nodes. The last two columns provide the number of each subgraph in the two signaling databases.

Simple Subgraphs
Structures	Names	Abbrev.	KEGG	OmniPath
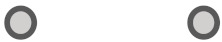	Unconnected Gene Pairs	UGP	__	__
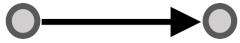	Activation	Act	19,170	15,841
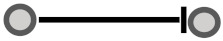	Inhibition	Inh	7320	5012
Complex Subgraphs
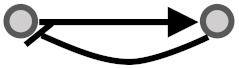	Dual Negative Feedback Loop	DNFBL	37	279
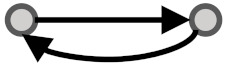	Dual Positive Feedback Loop1	DPFBL1	186	912
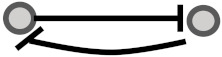	Dual Positive Feedback Loop2	DPFBL2	14	173
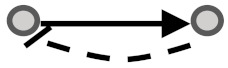	Multiple Negative Feedback Loop1	MNFBL1	17,712	14,913
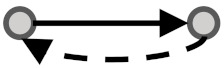	Multiple Positive Feedback Loop1	MPFBL1	3731	4104
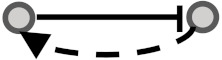	Multiple Negative Feedback Loop2	MNFBL2	2417	1005
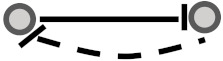	Multiple Positive Feedback Loop2	MPFBL2	3232	3279
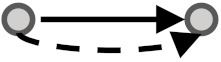	Multiple Feed-Forward Loop1	MFFL1	12,869	6729
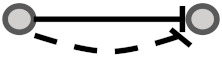	Multiple Feed-Forward Loop2	MFFL2	6618	4718
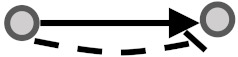	Multiple Negative Feed-Forward Loop1	MNFFL1	8918	9663
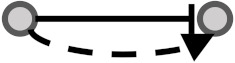	Multiple Negative Feed-Forward Loop2	MNFFL2	2925	842

**Table 2 biomolecules-10-00850-t002:** General properties and date retrieved of the signaling networks. The number of differentially expressed genes (DEGs) is also given, which are those common between the edge list genes and gene expression profile genes and identified by the GEO/GDSC database as either up- or down-regulated. Samples are the total number of samples in GEO and GDSC databases for which expression data were available for the given gene pair. The node number of the giant component, the diameter of the network, and the ratio of shared genes between edge list genes and gene-expression-profile genes are presented in the last three columns.

	KEGG						OmniPath					
	Date Retrieved	DEGs	Samples	Giant Component	Diameter	Ratio	Date Retrieved	DEGs	Samples	Giant Component	Diameter	Ratio
GEO	2017.08	3047	40,903	2549	17	0.95	201,905	4724	40,774	3848	17	0.95
GDSC	2017.10	2745	1018	2583	17	0.16	201,905	4402	1018	4045	15	0.25

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
