# Peer review of "Can We Assume the Gene Expression Profile as a Proxy for Signaling Network Activity?"

_biomolecules, 2020, doi:10.3390/biom10060850_

Round 1
Reviewer 1 Report
Manuscript of Piran and colleagues, entitled: "Can we Assume the Gene Expression Profile as a Proxy for Signaling Network Activity?" presents interesting issues in the field of systems biology and computational biology. It perfectly fits into the current of intensively conducted research in the field of genomics, proteomics and translational research, using high-throughput technologies.
The work is well written. The aim of the work is properly formulated and achieved. The methods of analysis are clearly described and detailed, the results obtained are presented visually in an interesting way and correspond to the assumed purpose. On their basis, the authors drew important conclusions, making a significant contribution to looking at the interpretation of gene expression data. A comparative analysis was carried out and adequate statistical tests were used. I have no substantive comments. The Figures are aesthetic, although their quality should be improved (they are hardly legible). The same remark applies to the tables (I think it should be reformatted). I have no further comments on the text.
Author Response
Responses to Reviewer’s Comments
Comments of Reviewer #1:
Manuscript of Piran and colleagues, entitled: "Can we Assume the Gene Expression Profile as a Proxy for Signaling Network Activity?" presents interesting issues in the field of systems biology and computational biology. It perfectly fits into the current of intensively conducted research in the field of genomics, proteomics and translational research, using high-throughput technologies.
The work is well written. The aim of the work is properly formulated and achieved. The methods of analysis are clearly described and detailed, the results obtained are presented visually in an interesting way and correspond to the assumed purpose. On their basis, the authors drew important conclusions, making a significant contribution to looking at the interpretation of gene expression data. A comparative analysis was carried out and adequate statistical tests were used. I have no substantive comments. The Figures are aesthetic, although their quality should be improved (they are hardly legible). The same remark applies to the tables (I think it should be reformatted). I have no further comments on the text.
Responses for Reviewer #1:
We are very appreciative of the honorable Reviewer #1’s encouragement and useful recommendations for this manuscript. We thank Reviewer #1 for pointing out the problem of reading Figures. We have now improved the font size of the figures to be more readable. We hope it now fulfills the requirements as instructed by Reviewer #1.
Reviewer 2 Report
The authors are interested in the coherence between the expression levels of genes forming gene pairs in human signaling networks. To analyze coherence they used four experimentally established data sets, two sets contain gene expression data and two sets are signaling pathway databases. The authors formed signaling/expression pairs by independently using two pathway databases to reconstruct a whole signaling network (for each database), and combining them with the gene expression databases to apply the correlation analysis on each gene pair in pathways they extracted. They compared the four thus combined data sets (KEGG/GEO, KEGG/GDSC, OmniPath/GEO, and OmniPath/ GDSC). Their aim was ”to evaluate the ratio of coherency or sign consistency between the expression level and the causal relationships among the gene pairs.” They found that “the signaling network was inconsistent or incoherent with the recorded expression profile compared to the random gene pairs.” They also “showed evidence and “concluded that for inferring disconcerted signaling pathways … logical relationships in addition to gene-product expression data should be used also, especially at the transcript level.” They ” asserted that identifying differentially expressed genes and enriched biological processes showed the limitation for presuming the existence of dysregulated pathways.”
Although the authors’ conclusion, based only on the data available to them, is convincing, there is a number of issues that should be addressed, in particular in relation to their conclusion.
- Besides proteins, noncoding RNA segments have been found to have enzymatic/controlling roles. The authors should discuss that in detail because noncoding (miRNA, siRNA) have very important roles.
- Alternative splicing of genes has not been addressed and it is known that in higher organisms it strongly affects sequences and therefore protein products produced by genes.
- The authors draw their conclusions using the analysis on the whole sets, or on randomly generated pairs, but signaling networks strongly depend on cells, tissues and organisms. Can they address signaling pathways and gene pairs in specific tissues or diseases (cancer). Maybe gene coherence analysis produces stronger signals when the analysis is focused.
- Can they explain the sentence “The important point is that all possible sort of relationships has been considered intrinsically” they used in Discussion. The sentence is not grammatically correct.
- The sentence ““Incoherency in signaling networks at transcript level would be a beneficial tool for biological systems to become adapted to both genetic and environmental changes” implies that gene signaling networks and therefore cells are robust, and that signals in cells not amplitude modulated, but perhaps frequency modulated. That should not be the only case in nature (e.g. some nerve signals are frequency modulated).
- The authors only emphasize phosphorylation as posttranslational modification. They should mention other such modifications (methylation, acetylation, ubiquitylation, sumoylation, biotinylation, etc).
- The authors should maybe mention that their analysis shows that maybe the four common databases do not contain all the relevant data.
Minor details:
- Sentence in abstract: “However, investigating the relation of the logical interactions of gene products to their expression profiles has not widely studied.” The sentence should be “However, investigating the relation of the logical interactions of gene products to their expression profiles has not been widely studied.” The two sentences after that sentence should be corrected also :“We aimed to evaluate the ratio of coherency or sign consistency between the expression level and the causal relationships among the gene pairs. We illustrated that the signaling network was inconsistent or incoherent with the recorded expression profile compared to the random gene pairs. Finally, we provided the pieces of evidence and concluded that for inferring disconcerted signaling pathways, we need to consider the logical relationships in addition to gene-product expression data, especially at the transcript level. We asserted that identifying differentially expressed genes and enriched biological processes showed the limitation for presuming the existence of dysregulated pathways.”
- Abbreviation DEG is used before being defined!
- The manuscript should be thoroughly read before the next submission. The text is very difficult to follow and it seems it is not aimed at the general audience, but to a very specific subgroup of the research community. That limits its readership. For instance, the authors should give a detailed explanation what is coherence, and give the expression/equation used for coherence between two signals as used in telecommunications. They should, maybe, clearly mention that they were looking for causation derived correlation.
- There is a number of grammatically incorrect sentences that should be corrected.
Author Response
Responses to Reviewer’s Comments
Part II
Comments of Reviewer #2:
The authors are interested in the coherence between the expression levels of genes forming gene pairs in human signaling networks. To analyze coherence they used four experimentally established data sets, two sets contain gene expression data and two sets are signaling pathway databases. The authors formed signaling/expression pairs by independently using two pathway databases to reconstruct a whole signaling network (for each database), and combining them with the gene expression databases to apply the correlation analysis on each gene pair in pathways they extracted. They compared the four thus combined data sets (KEGG/GEO, KEGG/GDSC, OmniPath/GEO, and OmniPath/ GDSC). Their aim was ”to evaluate the ratio of coherency or sign consistency between the expression level and the causal relationships among the gene pairs.” They found that “the signaling network was inconsistent or incoherent with the recorded expression profile compared to the random gene pairs.” They also “showed evidence and “concluded that for inferring disconcerted signaling pathways … logical relationships in addition to gene-product expression data should be used also, especially at the transcript level.” They ” asserted that identifying differentially expressed genes and enriched biological processes showed the limitation for presuming the existence of dysregulated pathways.”
Although the authors’ conclusion, based only on the data available to them, is convincing, there is a number of issues that should be addressed, in particular in relation to their conclusion.
- Besides proteins, noncoding RNA segments have been found to have enzymatic/controlling roles. The authors should discuss that in detail because noncoding (miRNA, siRNA) have very important roles.
- Alternative splicing of genes has not been addressed and it is known that in higher organisms it strongly affects sequences and therefore protein products produced by genes.
- The authors draw their conclusions using the analysis on the whole sets, or on randomly generated pairs, but signaling networks strongly depend on cells, tissues and organisms. Can they address signaling pathways and gene pairs in specific tissues or diseases (cancer). Maybe gene coherence analysis produces stronger signals when the analysis is focused.
- Can they explain the sentence “The important point is that all possible sort of relationships has been considered intrinsically” they used in Discussion. The sentence is not grammatically correct.
- The sentence ““Incoherency in signaling networks at transcript level would be a beneficial tool for biological systems to become adapted to both genetic and environmental changes” implies that gene signaling networks and therefore cells are robust, and that signals in cells not amplitude modulated, but perhaps frequency modulated. That should not be the only case in nature (e.g. some nerve signals are frequency modulated).
- The authors only emphasize phosphorylation as posttranslational modification. They should mention other such modifications (methylation, acetylation, ubiquitylation, sumoylation, biotinylation, etc).
- The authors should maybe mention that their analysis shows that maybe the four common databases do not contain all the relevant data.
Minor details:
- Sentence in abstract: “However, investigating the relation of the logical interactions of gene products to their expression profiles has not widely studied.” The sentence should be “However, investigating the relation of the logical interactions of gene products to their expression profiles has not been widely studied.” The two sentences after that sentence should be corrected also: “We aimed to evaluate the ratio of coherency or sign consistency between the expression level and the causal relationships among the gene pairs. We illustrated that the signaling network was inconsistent or incoherent with the recorded expression profile compared to the random gene pairs. Finally, we provided the pieces of evidence and concluded that for inferring disconcerted signaling pathways, we need to consider the logical relationships in addition to gene-product expression data, especially at the transcript level. We asserted that identifying differentially expressed genes and enriched biological processes showed the limitation for presuming the existence of dysregulated pathways.”
- Abbreviation DEG is used before being defined!
- The manuscript should be thoroughly read before the next submission. The text is very difficult to follow and it seems it is not aimed at the general audience, but to a very specific subgroup of the research community. That limits its readership. For instance, the authors should give a detailed explanation what is coherence, and give the expression/equation used for coherence between two signals as used in telecommunications. They should, maybe, clearly mention that they were looking for causation derived correlation.
- There is a number of grammatically incorrect sentences that should be corrected.
Responses for Reviewer #2.
We sincerely show our great appreciation for Reviewer #2’s comments. These comments are useful to improve the readability of our manuscript. We have now revised according to each comment or concern point by point below. We hope our responses fulfill the requirements.
Reviewer#2, Major Concern # 1: Besides proteins, noncoding RNA segments have been found to have enzymatic/controlling roles. The authors should discuss that in detail because noncoding (miRNA, siRNA) have very important roles.
Author response: Thank you for pointing this out about the biological systems. Analyzing non-coding RNA is out of the scope of the current manuscript since our focus is only on signaling networks. These networks are enriched by proteins and metabolites, and we checked the expression profiles of each gene-product (protein) pair to compare with their logical structure. Although the noncoding RNA affect signaling events by regulating the expression of proteins, they do not have a major role in the signaling pathways as a signaling component [Wang, K.C. and Chang, H.Y., 2011. Molecular mechanisms of long noncoding RNAs. Molecular cell, 43(6), pp.904-914.]. So, most of the interactions between noncoding RNA and proteins are not part of signaling networks, but gene regulatory networks which are not our focus. However, the concerns indeed benefited the Discussion part of the manuscript to show this distinction (Page 11).
Author action: We added the following to the Discussion:
“Although the regulation of gene expression results in a particular concentration of proteins, it is not sufficient to completely describe protein abundances (31, 32). The roles of other mechanisms, such as post-transcriptional, post-translational, and protein degradation regulations, have been reported to control steady-state protein abundance and protein activity (31, 33). For example, miRNAs are associated with proteins and diseases as regulators of gene expression (34, 35). There are several studies focusing on the regulatory role of long noncoding RNAs (lncRNAs) in wound healing and cancers (36-38).”
Reviewer#2, Major Concern # 2: Alternative splicing of genes has not been addressed and it is known that in higher organisms it strongly affects sequences and therefore protein products produced by genes.
Author response: Thank you for this valuable comment. Alternative splicing of genes strongly affects sequences and the final protein products; however, most of the available microarray gene expression datasets in GEO and GDSC databases do not include the information of different spliced variants of genes. We extracted the protein names involved in the signaling networks, and then we tried to find the expression profiles of proteins by matching gene names among the gene expression datasets. Moreover, the gene expression dataset mostly contains the expression profiles of mature mRNA, and all post-transcriptional processes, including alternative splicing, are inherently considered in expression datasets.
Author action: We have now added the following to the Discussion part of our manuscript at page 11:
“Alternative splicing also results in diverse forms of proteins, and the functions of proteins and corresponding signal transduction pathways are affected by this type of regulation in some diseases (39)”
Reviewer#2, Major Concern # 3: The authors draw their conclusions using the analysis on the whole sets, or on randomly generated pairs, but signaling networks strongly depend on cells, tissues and organisms. Can they address signaling pathways and gene pairs in specific tissues or diseases (cancer). Maybe gene coherence analysis produces stronger signals when the analysis is focused.
Author response: Thank you for this comment. First of all, we should highlight that we separated datasets in each repository to avoid the issue of heterogeneity of different organisms, tissues, and cells. Then we analyzed the correlations independently in each dataset. In Figure 2A, we tried to represent the effect and importance of this preprocessing. So, in this study, the gene expression profiles considered were dataset-specific. Additionally, we added GDSC datasets to our Pipeline with the same purpose. The GDSC database is the largest public repository that archives information about drug sensitivity in cancer cell lines. As expected, the coherency is improved in comparison with the GEO, see figure 3B in which the percent of coherent edges increased to 28.1% (13.5% in GEO) and 34.2% (11.1% in GEO) compared to GEO respectively. However, we clarified this issue and our solution approach more clearly within the revised manuscript.
Author action: We tried to explain this matter more clearly in the following sections at Pages 8 and 9:
“In addition, the ratio of eligible edges was higher in GDSC compared with GEO. This comparison may indicate the higher quality of gathered and annotated data in OmniPath and GDSC databases.”
“The ratio of coherent gene pairs in OmniPath was, in general, higher than for KEGG, while the ratio of coherent gene pairs in the GDSC database was higher than that of GEO. In detail, when we considered GDSC as a cancer-specific homogenous dataset, the percent of coherent edges increased from 13.5% in GEO to 28.1% according to the KEGG database and from 11.1% in GEO to 34.2% based on the OmniPath database.”
Reviewer#2, Major Concern # 4: Can they explain the sentence “The important point is that all possible sort of relationships has been considered intrinsically” they used in Discussion. The sentence is not grammatically correct.
Author response: We are grateful for pointing out this improper use of words in this sentence.
Author action: We removed this sentence since it does not add critical and essential information.
Reviewer#2, Major Concern # 5: The sentence ““Incoherency in signaling networks at transcript level would be a beneficial tool for biological systems to become adapted to both genetic and environmental changes” implies that gene signaling networks and therefore cells are robust, and that signals in cells not amplitude modulated, but perhaps frequency modulated. That should not be the only case in nature (e.g. some nerve signals are frequency modulated).
Author response: Thank you for your useful comment to edit this sentence.
Author action: We revised the sentence as follows at page 11:
“Incoherency in signaling networks at the transcript level might be a strategy used by cells to tune both genetic and environmental signals or a byproduct resulting from frequent modulation of signal transduction within biological systems.”
Reviewer#2, Major Concern # 6: The authors only emphasize phosphorylation as posttranslational modification. They should mention other such modifications (methylation, acetylation, ubiquitylation, sumoylation, biotinylation, etc).
Author response: Thanks for pointing this out. We mentioned some other modifications with the corresponding references in several parts of the revised manuscript.
Author action: We have now added the following sentences and related references at pages 3 and 11:
“Moreover, numerous methylation sites have been discovered on human proteins that impact cell signaling (40).”
“When considering signaling networks as a portrait of causal relationships among the molecular entities in biology, their leading players are proteins, whose activity is often regulated by post-translational modifications such as phosphorylation and acetylation (9, 10).”
“These unexplained variations might come from organism-specific translational and post-translational regulations, including protein degradation by ubiquitylation and sumoylation (29).”
Reviewer#2, Major Concern # 7: The authors should maybe mention that their analysis shows that maybe the four common databases do not contain all the relevant data.
Author response: Thanks for this suggestion. We agree to add this point to not generalize our discussion to all datasets.
Author action: We added the following sentences in the conclusion at page 13:
“Note that these observations are according to the four datasets used here, and independent investigation of gene pair relationships based on more accurate and specific datasets is recommended.in”
Reviewer#2, Minor Concern # 1: Sentence in abstract: “However, investigating the relation of the logical interactions of gene products to their expression profiles has not widely studied.” The sentence should be “However, investigating the relation of the logical interactions of gene products to their expression profiles has not been widely studied.” The two sentences after that sentence should be corrected also: “We aimed to evaluate the ratio of coherency or sign consistency between the expression level and the causal relationships among the gene pairs. We illustrated that the signaling network was inconsistent or incoherent with the recorded expression profile compared to the random gene pairs. Finally, we provided the pieces of evidence and concluded that for inferring disconcerted signaling pathways, we need to consider the logical relationships in addition to gene-product expression data, especially at the transcript level. We asserted that identifying differentially expressed genes and enriched biological processes showed the limitation for presuming the existence of dysregulated pathways.”
Author response: Thank you for your comment. The mentioned sentences are now revised in the current manuscript.
Reviewer#2, Minor Concern # 2: Abbreviation DEG is used before being defined!
Author response: Thank you for your consideration. We now provide this abbreviation in its first mention.
Reviewer#2, Minor Concern # 3: The manuscript should be thoroughly read before the next submission. The text is very difficult to follow and it seems it is not aimed at the general audience, but to a very specific subgroup of the research community. That limits its readership. For instance, the authors should give a detailed explanation what is coherence, and give the expression/equation used for coherence between two signals as used in telecommunications. They should, maybe, clearly mention that they were looking for causation derived correlation.
Author response: We thank you for pointing out this issue. We now have revised the whole manuscript to fulfill this requirement, and additionally the coherence concept is further clarified at pages 3 and 4 as follows:
“In this study, we aimed to examine the coherency between expression profiles and the signs of relationship in signaling networks for all possible gene pairs. To elaborate on what we mean by coherency, imagine a gene pair (GI, GII) where gene GI activates gene GII. If the expression profiles of both are correlated positively, we infer that expression data strengthen the activation type of this signaling relationship and are thus a coherent gene pair. By contrast, let gene GI inhibit gene GII. In this case, the coherent gene pairs are negatively correlated, and the inhibition type of this signaling relationship is strengthened in accordance with the expression levels of transcripts. If gene GI activates gene GII while there is a negative correlation between them or if gene GI inhibits gene GII while there is a positive correlation between them, this implies the incoherency between the gene pair relationships. In other words, incoherent gene pairs’ effects on the action of each other are not supported by their mRNA expression levels. In addition to these simple scenarios, we also considered more complicated subgraphs in a signaling network (See Table 1). Edges appear between gene pairs, indicating a relationship. Since all edges have a direction, the type of logical relationship is determined using the sign of edges. We then performed correlation analysis between the connected gene pairs compared with unconnected gene pairs to infer coherency.”
Reviewer#2, Minor Concern # 4: There is a number of grammatically incorrect sentences that should be corrected.
Author response: Thank you for your comment. The current version is now edited.